# A Review on the Synthesis of Fluorescent Five- and Six-Membered Ring Azaheterocycles

**DOI:** 10.3390/molecules27196321

**Published:** 2022-09-25

**Authors:** Gheorghita Zbancioc, Ionel I. Mangalagiu, Costel Moldoveanu

**Affiliations:** 1Chemistry Department, Alexandru Ioan Cuza University of Iasi, 11 Carol 1st Bvd., 700506 Iasi, Romania; 2Institute of Interdisciplinary Research-CERNESIM Centre, Alexandru Ioan Cuza University of Iasi, 11 Carol I, 700506 Iasi, Romania

**Keywords:** five and six membered ring azaheterocycles, fluorescence, optical properties, synthesis

## Abstract

Azaheterocycles rings with five and six members are important tools for the obtaining of fluorescent materials and fluorescent sensors. The relevant advances in the synthesis of azaheterocyclic derivatives and their optical properties investigation, particularly in the last ten years, was our main objective on this review. The review is organized according to the size of the azaheterocycle ring, 5-, 6-membered and fused ring azaheterocycles, as well as our recent contribution on this research field. In each case, the reaction pathways with reaction condition and obtained yield, and evaluation of the optical properties of the obtained products were briefly presented.

## 1. Introduction

The synthesis of fluorescent azaheterocycles continues to arouse strong interest due to their great potential for application as sensors and biosensors, luminophores on in the construction of Organic Light-Emitting Diode (OLED) devices, laser and other semiconductor devices [1,2,3,4], as well as to their potential biological properties as antimicrobial [5], antifungal [6], anticancer [7,8], antituberculosis [9,10] antioxidant [11] and anti-HIV [12] agents.

The advantages of the azaheterocyclic fluorophores, such as small size, enriched photostability, a wide and tunable spectral range, and, frequently, high brightness, are the reason why these fluorophores are preferred and used in various medical application. Probe structure can be modified to adjust excitation and emission wavelengths, target-binding affinity, chemical reactivity, and subcellular localization [13,14,15,16].

In this review, we try to present an overview of the newest research in the synthesis of fluorescent azaheterocyclic derivatives, from the literature reported during the last ten years. The methodology adopted for literature search and selection was to access the Reaxys database for “fluorescence”, “azaheterocycles” and “fluorescent azaheterocycles” keywords. The obtained results, as well as the references cited, were selected to be included in the present review if they present the synthesis of fluorescent azaheterocycles, and the fluorescent properties of these compounds were evaluated.

The present review was organized taking into account the size of the azaheterocycle, the nature of the cycle (monocycle and fused polycycles), as well as our contribution in this research field.

## 2. Synthesis and Fluorescent Properties of Five- and Six-Membered Ring Azaheterocycles

### 2.1. Five Membered Ring Azaheterocycles

Five membered ring azaheterocycles, such as pyrroles, diazoles, triazoles, tetrazoles and their fused derivatives, are privileged scaffolds posing a wide range of biological activities and diverse applications in organic electronics [17,18,19,20,21,22,23,24,25]. Due to these properties, many researchers were focused on the obtaining of new compounds with such skeleton.

Martínez-Lara et al. [26] developed a new obtaining methodology of two different angular indolocarbazole moieties by using two sequential gold- and molybdenum-catalysis to obtain new indolo [2,3-*c*]carbazoles as potential luminophores with application in OLED devices.

A selection of 1,1-bis(indol-3-yl)-3-alkyn-2-ols **3**, was readily prepared through the alkylation of the indole derivatives **1** with methyl or phenyl glyoxal, followed by a standard Sonogashira-type reaction with 2-nitroaryl iodides. Next, these compounds underwent a cyclization reaction, promoted by NaAuCl_4_, affording the corresponding carbazoles **4** (see Figure 1).

Without further purification, the crude carbazoles were directly subjected to Mo-catalyzed Cadogan reaction delivering the desired indolocarbazoles **5**. All indolocarbazoles were obtained in high yields. The authors [26] used microwave irradiation for the synthesis of the indolocarbazole derivatives **4** and **5**, and obtained final products in much shorter time and with better yields (see Figure 2).

To obtain a perspective on the effect that the C-7 substituent of the indolocarbazole skeleton (methyl in **5aa**, and phenyl in **5ba**) the photoluminescent properties of two selected compounds, **5aa** and **5ba** in DMSO were investigated Table 1.

The phenyl-substituted **5ba** indolo[2,3-*c*]carbazole shows a wider absorption wavelength range than methylsubstituted **5aa** indolo[2,3-*c*]carbazole and both show fluorescence quantum yields around 70% in DMSO, double as the Φ_fl_ of some similar compounds from the literature. These compounds are potentially useful for optoelectronic applications due to their high extinction coefficient and fluorescence quantum yields.

By combining 2H-imiazole 1-oxides **7a**–**l** with pentafluorophenyl lithium **8** made in situ from pentafluorobenzene **6** and n-BuLi, Moseev et al. [27] were able to create novel pentafluoroaryl-modified derivatives of 2H-imidazole type **10a**–**l** and **11a**–**l**, as is presented in Figure 3. Pentafluorophenyllithium **8** first attacks the imidazole-N-oxide **7a**–**l**, resulting in the unstable σ^H^-adduct **9** that can be converted into the product either by “addition-elimination” (Path A), leading to the formation of compounds **10a**–**l**, or by “addition–oxidation” (Path B), leading to the creation of compounds **11a**–**l** containing N-oxide group.

As possible applications, the authors claim that the push-pull fluorophore systems containing both electron-donating (EDG) and electron-withdrawing (EWG) groups in the vicinal position could be used as push–pull fluorophore systems. The obtained molecule types **10** and **11** have undergone a photophysical study to determine the potential practical uses in the construction of photoactive materials. The authors results are shown in Table 2.

The authors concluded that the presence of EWG in the *para*-position of the phenyl group improved the fluorescence quantum yield of the produced fluorophores by measuring the quantum yield of polyfluorinated-2Himidazole derivatives **10** and **11**.

The phenomenon of intramolecular charge transfer (ICT) in polar protonic solvents (MeOH and EtOH) has been shown for several compounds, and connections between “structure-photophysical characteristics” have been found. The characteristics of the investigated photophysical properties allow them to adjust the structures of the photoactive molecules, which opens the possibility of using the synthesized push-pull systems in the development of difficult fluorometric molecular sensors.

Pu et al. [28] synthesized antipyrine-containing diarylethenes derivatives **14** in their attempt to obtain novel multi-controllable photochromic and fluorescent derivatives (see Figure 4).

It was determined that **14o** had an absolute fluorescence quantum yield of 1.4%. UV light (297 nm) irradiation caused the photocyclization reaction, which resulted in the synthesis of **14c** with a quantum yield of 0.8%. As a result, the emission strength of **14o** abruptly reduced, and the fluorescence changed from bright orange to darkness. Conversely, by exposing **14o** with appropriate visible light, the fluorescence may be recovered.

At 580 nm, the isomer **14o** showed high brightly orange fluorescence. The emission strength of **14o** reduced to 18% with the addition of 5.0 equiv of tetrabutylammonium hydroxide, and the color changed noticeably from bright orange to blackness as a result of the creation of deprotonated **14o’**. The fluorescence could return to its initial form upon neutralization with 2.0 equiv of HCl. Similar to this, the stimulation of TBAH/HCl in acetonitrile could reversibly alter the fluorescence of **14c**. When irradiated to UV/vis light, the fluorescence alterations between **14o’** and **14c’** was irreversible. The fluorescence of **14o’** stayed unaltered when irradiated to UV light, however when exposed to visible light (λ > 500 nm), the fluorescence of **14c’** might return to that of **14o’**, as may be observed in Figure 5. The results obtained by the authors shown that the base could suppress the photochromism of diarylethene with an antipyrine unit’s and acid could restore it.

In order to examine the fluorescence variations of **14o**, 5.0 equiv of several metal ions, including Al^3+^, Zn^2+^, Ni^2+^, K^+^, Cd^2+^, Ca^2+^, Cu^2+^, Ba^2+^, Pb^2+^, Cr^3+^, Co^2+^, Sr^2+^, Mg^2+^, Mn^2+^, Hg^2+^ and Fe^3+^, were added. The authors pointed out that the addition of Al^3+^ was the only metal ion that significantly affected the fluorescence of **14o**; other metal ions had a negligible effect. As a result of this study, the diarylethene with an antipyrine unit type **14o** demonstrated special base-gated photochromic properties and could be used as a highly selective naked-eye chemosensor for Al^3+^ detection.

### 2.2. Six Membered Ring Azaheterocycles

Electron-deficient six membered ring azaheterocycles, such as azines and diazines, are strongly desired and studied due to their potential biological activities and also to their potential applications in organic electronics [29,30,31,32,33].

Motivated by the small size and high quantum yield (Φ = 60%) of the unsubstituted pyridin-2-amine, being a potential scaffold for a fluorescent probe, Li et al. [34], in their research for new multisubstituted aminopyridines, used the Rh-catalyzed coupling of vinyl azide with isonitrile to form a vinyl carbodiimide intermediate, which followed a tandem cyclization with an alkyne to the desired amine.

The reactions were carried “*one-pot*”, using vinyl azide **15** and isonitrile **16**, Rh-catalyst, proper ligand and 1,4-dioxane as solvent, under N_2_ atmosphere and at room temperature in the first stage, then after the vinyl azide spot disappeared on TLC, NH_4_Cl, NaHCO_3_, and alkyne were added, and the mixture was heated to 120 °C for 8 h (see Figure 6).

For quantitative fluorescent detection, the solution of aminopyridine was then diluted to 10 μM. The measured parameters are presented in Table 3.

Compound **17** was derivatized in order to observe the variation of the photoluminescent properties, as is presented in Figure 7.

Thus, when the tertiary butyl group was cleaved using CF_3_COOH the obtained compound **27** showed no fluorescence. When the ester groups were hydrolyzed to carboxylic acid, the obtained compound **28** still had a good fluorescence (Φ = 31%). When compound **35** was reduced with LiAlH_4_ a high fluorescent (Φ = 81%) alcohol **29** was obtained in excellent yield, but the λ_em_ was reduced to 400 nm.

The same authors [34], in order to apply the newly synthesized compound as fluorescent probe, used the click reaction to bind the fluorescent aminopyridine scaffold to biomolecules conjugated with alkynyl group (see Figure 8). Firstly, they synthesized a Boc-NH protected compound **30** accordingly to the above-described method from corresponding starting materials, then the azido substituted derivative **31** was obtained via a Sandmeyer reaction. Finally, the azide **31** react easily with phenylacetylene or propynol (by using catalytic amounts of Cu(II)) to obtain the desired triazole products **32** or **33**.

The authors successfully applied this “click-and-probing” experiment to bovine serum albumin (BSA) conjugated with a terminal alkyne, demonstrating the biochemical application of this probe and of the fluorescent multisubstituted aminopyridines for detection and analysis.

Piloto et al. [35] studied the use of acridine azaheterocycle as a photochemically removable protecting group for the carboxylic group of amino acids involved in neurotransmision. In this respect, a series of amino acids **34** (glycine, alanine, glutamic acid, β-alanine and γ-aminobutyric acid) protected on amino group with *tert*-butoxycarbonyl, was treated with 9-bromomethylacridine **35**, the corresponding heterocyclic ester **36** being obtained (see Figure 9). The coupling reaction requires potassium fluoride as base and N,N-dimethylformamide as solvent, and take place at room temperature.

UV/Vis absorption and emission spectra for the ester conjugates **36a**–**e** were measured on degassed 10^−5^ M solutions in two solvent systems: absolute ethanol, and a methanol–HEPES buffer (80:20), the measured parameters are presented in Table 4.

Due to the heterocyclic chromophore, the absorption maximum of compounds showed no influence of amino acid residue. Ester derivatives **36a**–**e** displayed emission maxima between 411–435 nm in both solvents, with moderate Stokes’ shifts (50–75 nm). These esters exhibited higher fluorescence quantum yields in a methanol–HEPES buffer solvent system.

The deprotection of the carboxylic group from the corresponding fluorescent derivatives in methanol-HEPES buffer (80:20) solutions was achieved by irradiation. The best results on photorelease of these amino acids from were obtained on irradiation at 350 nm. This ability of acridinyl methyl ester make them a suitable option for the photochemical release of functional molecules bearing a carboxylic acid group in organic synthesis and in cellular applications.

In order to obtain new luminogens for sensing strong acids Tang et al. [36] used Suzsuki cross-coupling reaction. The luminogens consist of one pyridine, 1,3-diazine, 1,4-diazine, 1,2-diazine and phthalazine moieties as the central cores and two AIE-active tetraphenylethene units in the lateral sides. Aggregation induced emission (AIE) effect exhibiting high emission in concentrated solution or even in the solid state, hold great promise for a luminogen in practical applications.

PY-TPE **39b**, PYM-TPE **39c**, PYA-TPE **39d**, PYD-TPE **39e** and PTZ-TPE **39f** were easily synthesized by 4-(1,2,2-triphenylvinyl) phenyl boronic acid **37** reaction with 2,5-dibromopyridine **38b**, 2,5-dibromopyrimidine **38c**, 2,5-dibromopyrazine **38d**, 3,6-dibromopyridazine **38e**, 1,4-dibromophthalazine **38f** under Suzuki cross-coupling in high yields (see Figure 10). For comparison, an analogue molecule BZ-TPE **38a** fused with phenyl unit was also prepared.

The photophysical properties of the luminogens were studied by UV-vis and fluorescence spectroscopies in CH_3_CN, and are presented in Table 5. These luminogens show a colorless character in common organic solvents, making them good candidates for colorimetric or fluorometric detection with less self-absorption disturbance. All compounds show very weak fluorescence emission in CH_3_CN, caused by non-radiative fashion arising from the rotation of tetraphenylethene units.

The emission spectra (in CH_3_CN–water) were recorded in order to determine the AIE effect in aggregate states of the as-prepared luminogens. Benzyl-, pyridine-, pyrimidine- and pyrazine-tetraphenylethenes (**39a**–**d**) exhibited high fluorescence quantum yields in aggregate states due to a typical AIE effect. 1,2-Diazine-tetraphenylethenes (pyridazine-**39e** and phthalazine-**39f**) have strong yellow fluorescence emission under protonation enabling AIE performance. The synergetic effect of AIE and the protonation on the 1, 2-diazine segments allows them to detect strong acids with very low pKa values.

The similarities of the 4-hydroxy-1,3-thiazole unit, a chromophore and fluorophore, with naturally occurring luciferin, determine Menzel et al. [37] to obtain new of arylamine-modified thiazoles as donor-acceptor dyes. The donor part of the dyes was based on the 4-methoxy-1,3-thiazole core substituted with an arylamine (phenyl-, *p*-anisole-, *p*-tolyl-, or phenothiazine-) in the 5-position as donor, and a pyridine, pyrimidine, or pyrazine moiety in the 2-position as acceptor.

The synthesis of the 4-hydroxy-1,3-thiazoles **42a**–**c** involved a Hantzsch thiazole cyclizations between the corresponding azaheterocyclic thioamides **40a**–**c** and ethyl 2-bromo-2-(4-nitrophenyl)acetate **41a**. Similarly, starting from ethyl 2-bromo-2-(4-bromophenyl)acetate **41b** and pyridine-2-carbothioamide **40a**, compound **42d** was prepared, then methylated via Williamson ether synthesis, with methyl iodide in DMSO. The reduction of the nitro group in **43a**–**c** with freshly prepared Raney nickel and hydrazine in EtOH fallowed (see Figure 11).

The reaction condition for the double N-arylation Buchwald–Hartwig cross-coupling reaction were: bis(dibenzylideneacetone)palladium(0) [Pd(dba)_2_] as the precatalyst, KO*t*Bu as the base, and toluene as solvent. By using the P(*t*Bu)_3_ reagent, the desired products were obtained in moderate to good yields (76–90%). The obtaining of both disubstituted **A1** and **B1**, and the monosubstituted products **A1m** (69%) and **B1m** (88%), demonstrates the two-step nature of the reaction (see Figure 12).

The Buchwald–Hartwig reaction starting with **43d**, a thiazole substituted aryl halide, was not successful under the conditions of the double N-arylation of the amines **44a**–**c**, the biarylphosphane 2-(dicyclohexylphosphanyl)-2′,6′-dimethoxy-1,1′-biphenyl (SPHOS) was required as ligand (see Figure 13).

The arylamine donors has a strong influence on the emission quantum yields recorded in CH_3_CN solution at room temp, the phenyl-based triarylamines having good quantum yields (40–47%), while the *p*-anisole-based triarylamines show values below 1% for quantum yields, and consequently no fluorescence for the p-N,N′-dimethylaniline derivative. The obtained photoluminescence quantum yields (see Table 6) are correlated with the measured emission lifetimes (τ).

Only the emission due to the carbazole moiety, as a main band, can be observed for compound **D1** in the polar CH_3_CN, while for **D2** and **D3** the emissions are modest.

The authors [37] carried out measurements of the absorptions (and emissions) in different solvents for compounds **A2** and **D1** in order to investigate them in more detail (see Table 7). The increasing of the solvent polarity induces a weak hypsochromic shift on the absorption maxima, but a bathochromic shift on the emission maxima. This behavior is due to a conformational transformation from a planar locally excited (LE) state to an intramolecular charge-transfer (ICT) or, consequently, to a twisted intramolecular charge-transfer (TICT) state.

Menzel et al. [37] tested also the ability of these dyes to act as ligands and synthesized seven new ruthenium complexes and measured their emission spectra. The synthesis of the heteroleptic Ru^II^ complexes involve the activation of the cis-(dmbpy)_2_RuCl_2_ precursor with AgPF_6_ prior to the complexation in acetone with the appropriate ligand (1 equiv.) for 24 h under reflux conditions (see Figure 14).

Each complex shows an enhanced absorption in the visible part of the UV/Vis spectrum, due to additional ligand-centered (LC) π–π* transitions and metal-to-ligand charge-transfer (MLCT) transitions originating from 4-methoxy-1,3-thiazole ligands.

### 2.3. Fused Azaheterocycles

Fused azaheterocycles combine an electron-excessive pyrrole and an electron deficient azine ring which involve an uneven π-electron distribution. This uneven π-electron distribution induces interesting optical properties on the compound with such fused azaheterocycles and make them very attractive materials in optoelectronics [38,39]. Moreover, the variety of potential applications in the fields of medicinal of these compounds explains the increased interest of researchers in their study [40,41,42]. 

Rajbongshi et al. [38] obtained new chromeno[2,3-*b*]indoles **47a**–**c** as part of the series of chromenoindole derivatives by reacting 1-(α-amino-α-arylalkyl)-2-naphthol **45** with indole **46** at 100 °C in the presence of catalytic *p*-toluenesulfonic acid, I_2_ and tert-butyl hydroperoxide (TBHP), as is presented in Figure 15.

In various solvents, the fluorescence spectra of the produced chromeno[2,3-*b*]indoles **47a**–**c** alter and show a significant Stokes shift, ranging from 250 nm in acetonitrile to 186 nm in ethyl acetate. The chromeno[2,3-*b*]indoles’ fluorescence quantum yield was evaluated at 280 nm using tryptophan in water as a reference and at 313 nm using naphthalene. Table 8 displays the fluorescence quantum yields of obtained compounds.

The investigated chromeno[2,3-*b*]indoles **47a**–**c** produce fluorescence at a significantly lower energy compared to their absorption, and their fluorescence spectra exhibit a solvatochromic shift, which suggests intramolecular charge transfer in the excited state. The three chromeno[2,3-*b*]indoles’ with strong Stokes-shifted fluorescence suggests that they may be used as luminescent solar concentrators or scintillators.

Li et al. [39] obtained a fused five-membered azaheterocycle with an aggregation-induced emission (AIE) characteristic using an unanticipated regioselective photoreaction.

The starting compound *o*-TPBQ **48** were easily prepared through a facile one-step synthetic route via a modified Sonogashira coupling reaction according to the previously reported literature [43,44].

The authors proposed all of the potential products, including the common six-membered ones, **50** C_6_-TPBQ′ and **51** C_6_-TPBQ″, as well as the five-membered ring product **49** (C_5_-TPBQ), although the likelihood of this is very low. They also considered the specific location of the N atom in *o*-TPBQ and the reported literature on the photoreaction. Surprisingly, the typical six-membered cyclized product (C_6_-TPBQ’ or C_6_-TPBQ”) was not obtained; instead, only the five-membered cyclized product **49**, C_5_-TPBQ, was obtained (see Figure 16). NMR and high-resolution mass spectroscopies were used to confirm the structure of C_5_-TPBQ.

In DMSO/water solutions with various water fractions, the photoluminescence spectra of C_5_-TPBQ were recorded. This compound has a low fluorescence quantum yield in DMSO solution (1.1%), and when water is gradually added to the solution, an increased photoluminescence signal is obtained. The AIE characteristic was demonstrated by the 110-fold increase in emission intensity in DMSO/water mixes with 99% water compared with DMSO solution.

This research provides a method for quickly and easily creating fused five-membered azaheterocyclic compounds with unique fluorescence properties. These compounds have a wide range of uses in the biological and optoelectronic domains.

Having in view the synthesis of new 1*H*-pyrazolo[3,4-*b*]quinoxaline (PQX) derivatives with specific photophysical properties as potential materials for optoelectronics, Wojtasik et al. [45] tried to improve the Cadogan synthesis of these compounds. The authors used triphenylphosphine as reducing agent, in dimethylacetamide (DMAc) at 240 °C, under microwave irradiation and obtained PQX derivatives in much shorter time and with better yields of the final product (see Figure 17).

The substrates **52a**–**g** were obtained by the coupling of the appropriate 2-iodonitrobenzene derivatives **54** with 1,3-disubstituted 5-aminopyrrazole **55** in the presence of palladium catalyst and BINAP (2,2′-bis(diphenylphosphino)-1,1′-binaphtyl), as is presented in Figure 18. By this reaction, authors obtained the best yields, despite the long reaction time.

The recorded emission spectra of compounds **53e**–**g** reveal an emission maximum between 436 nm and 483 nm, according to the nature of the substituents from the 6th position and the used solvent. These results are presented in the Table 9.

The introduction of the electron donor substituent (methyl group-**53e**) in the 6th position of 1,3-dimethyl-1H-pyrazolo[3,4-*b*]quinoxaline **53d** did not changed the emission properties comparing with the unsubstituted **53d** system, while the electron-accepting substituents (chlorine-**53f** and trifluoromethyl group-**53g**) shifted the fluorescence band twoards longer wavelenghts by 8 nm to 21 nm, respectively, and induced an increase of the quantum yield of the fluorescence.

In order to develop a new family of thermally activated delayed fluorescence (TADF) emitters, utilizing the donor-acceptor scaffold, Goya et al. [46] focused on the use of the electron-deficient azaaromatic scaffold in place of the usual dibenzo[*a*,*j*]phenazine (DBPHZ) unit. Organic compounds that can display TDAF are very intense studied as emitters for efficient OLEDs, since they can achieve theoretically 100% internal quantum efficiency (IQE) by harvesting electrically generated triplet excitation and convert into the emissive singlet excitations through reverse intersystem crossing (rISC).

Pyrido[2,3-*b*]pyrazine (PYPZ) moiety was selected as the electron-acceptor (A) unit, due to the both pyridine and pyrazine π-deficient heterocycles. When this electron deficient core is connected with an appropriate electron-donor (D), intramolecular charge-transfer (ICT) in the excited state will occur and CT states should become the first singlet excited state (S1). As donor part was selected dihydrophenazasiline (DHPHAzSi) since DHPHAzSi compounds are moderate electron donor.

The 7-bromo-2,3-diphenylpyrido[2,3-*b*]pyrazine **59** and 10-bromo-acenaphtho[1,2-*b*]pyrido[2,3-*e*]pyrazine **61** donors were bonded to the corresponding DHPHAzSi acceptors through the Pd-catalyzed Buchwald-Hartwig amination in good yields (see Figure 19).

The starting compounds **59** and **61** were prepared according to the literature procedures [47,48], through the condensation between 2,3-diamino-5-bromopyridine with benzyl and phenanthrene-9,10-dione, respectively.

The photophysical properties of compounds **56**–**58**, were investigated in a non-polar polymer matrix Zeonex^®^, and in small molecule hosts, 4,4′-bis(N-carbazolyl)-1,1′-biphenyl (CBP) and tris(4-(9H-carbazoyl-9-yl)phenyl)amine (TCTA) and are presented in Table 10.

Compounds **56**–**58** manifest emissions in two different time domains. The first component is the prompt fluorescence (PF) from the singlet excited state (S1), with a lifetime in the nanosecond time delay, and the second components are the delayed emissions from the high triplet excited state (T2) (as theoretical calculations revealed), decaying in micro to millisecond delay time.

Georgescu et al. [49,50,51] studied the synthesis and fluorescence of 3-aryl-7-benzoyl-pyrrolo[1,2-*c*]pyrimidines in order to determinate the influence of the chemical structure and of solvent polarity on their optical properties.

The authors used, for the synthesis of 3-biphenyl-pyrrolo[1,2-*c*]pyrimidines, a *one-pot*, three-component procedure [49,50], which involve a 1,3-dipolar cycloaddition reaction of a 4-biphenyl pyrimidinium-N-ylide with an activated alkyne **65** in 1,2-epoxybutane at reflux. The ylide is generated “*in situ*” from the corresponding pyrimidinium salts that was formed by the N-alkylation of the pyrimidine **63** with halogenoketone **64**. The advantage of performing the reaction in a *one-pot* three-component approach is the direct formation of the final aromatic compounds **66a**–**j**, avoiding the formation of dipyrimidino-pyrazinic inactivated products (see Figure 20).

The absorption and emission spectra of compounds **66a**–**j** were recorded in acetonitrile:chloroform (1:1) (3.5 * 10^−6^ mol/L) solutions. The fluorescence quantum yield has been calculated (Table 11) for all compounds **66a**–**j** using Equation (1) using quinine sulphate as standard.
(1)$$\Phi_{fl}=\Phi_{ref}\times \frac{I}{A}\frac{A_{ref}}{I_{Ref}}\times \frac{n}{n_{ref}}$$

In order to calculate the quantum yield (Φ_fl_), according to the Equation (1) we need to determine the maximum value of the absorbance at the emission wavelength λ_em2_, (A), area of the emission peak (I), and refractive index (n) for the solution of investigated compound, and quantum yield (Φ_ref_), maximum value of the absorbance at the emission wavelength λ_em2_ (A_ref_), area of the emission peak (I_ref_), and refractive index (n_ref_) for the standard solution (quinine sulphate), respectively.

Looking at the values from Table 11, it can be seen that two of the compounds (**66c** *p*-flouoro substituted and **66f** 3,4-dimethoxy substituted) have higher values of quantum yield. These higher values of quantum yield can be explained by a more extended π electron conjugated system in the case of this compound. Ethyl 3-(4-biphenylyl)-7-(3,4-dimethoxybenzoyl)pyrrolo[1,2-*c*]pyrimidine-5-carboxylate was found to have the highest quantum yield value (55%). These higher quantum yield values suggest that the studied compounds are promising candidates for fluorescent chemical sensors.

In view of future exploitation of pyrrolo[1,2-*c*]pyrimidine compounds in the field of bio-imaging investigations Georgescu et al. [51] synthesized a series of new pyrrolo[1,2-*c*]pyrimidine derivatives and studied their photophysical properties. The synthetic approach was similar with the one described above, but this time did not use the one pot strategy, but only by the 1,3-dipolar cycloaddition of the pyrimidinium ylides generated “*in situ*” from the corresponding pyrimidinium bromides **67** with the alkyne dipolarophiles **68** in 1,2-epoxybutane as reaction medium and acid scavenger (see Figure 21).

The photophysical properties of fused derivatives **69a**–**j** were investigated by using different sorts of solvents (see Table 12). In chloroform, compounds derived from the ethyl propiolate, **69a**–**d**, present better fluorescence yield than the compounds derived from the symmetric alkynes **69e**–**j**. The excitation, during the recording of the emission spectra, was performed at λ_abs2_ at which a higher intensity of the fluorescence was obtained. A main emission band is located in the blue region of the visible spectrum in 437–463 nm region.

The authors [51] investigate the influence of all R_1_, R_2_, R_3_ and R_4_ substituents on the fluorescence properties of the synthesized compounds and concluded that a substituent that determine the growth of conjugation inside the pyrrolo[1,2-*c*]pyrimidine fragment lead to an increase of the fluorescence, while a substituent that determine the decrease of conjugation inside the pyrrolo[1,2-*c*]pyrimidine fragment lead to decrease of the fluorescence.

Tomashenko et al. [52] obtained new pyrido[2,1-a]pyrrolo[3,2-c]isoquinoline **76a**–**c** heterocyclic system by using Pd to catalyze the intramolecular cyclization of 1-[1-benzyl-2-(2-bromophenyl)-1H-pyrrol-3-yl]pyridin-1-ium bromides **74a**–**c** (see Figure 22).

The obtained compounds **76a**–**c** have fluorescent properties in solutions, with quantum yields for **76c** in methanol reaching 81%. Photophysical data were studied in toluene, acetonitrile, dichloromethane (DCM) and methanol, respectively (Table 13). The obtained compounds display moderate to strong fluorescence in solution, giving minor variations in emission maxima position with changes in solvent polarity. The biggest effect was shown in methanol solutions and upon addition of proton donors to aprotic solvents.

### 2.4. Our Recent Contribution to the Field

Nitrogen ylide chemistry is a traditional research area in our group, professors Zugravescu and Petrovanu being the pioneer of this field [53]. Several compounds with complex azaheterocyclic skeleton were synthesized through the N-ylides. Many of these compounds present practical importance either to their biological activities, having antimicrobial [40,41], anticancer [42,54,55,56,57,58], antitubercular [58,59,60,61] or anti-leishmaniasis [62] activity, or to their optical properties (some of this compound presenting intense blue fluorescence) [63,64,65,66,67,68,69]. Here we will present our recent (last 12 years) achievements in the synthesis of fluorescent azaheterocyclic compound.

In one extensive work [63,64], we carried out a comprehensive investigation into the synthesis of fluorescent pyrrolodiazine (PD). All PD derivatives **78**–**82** were produced using a methodology which involve two steps. In the first stage, diazinium salts **78a**–**h** were obtained by quaternization of diazine [pyridazine (PY) or phthalazine (PH)] with halogenated derivatives with increased reactivity. In the 2nd stage, a typical Huisgen [3+2] dipolar cycloaddition of diazinium ylides **78′** to the corresponding dipolarophiles were performed (see Figure 23).

The synthesis of the desired pyrrolodiazine derivatives **79**–**82** was performed under conventional thermal heating (TH) and using MW irradiation, by a typical Huisgen [3+2] dipolar cycloaddition. The reactions under MW irradiation have shown some important advantages such as significantly higher yields, decreasing the reaction time from hours to 5 min and five times less amount of solvent is needed.

To obtain a perspective on the effect that the C-7 substituent of the pyrrolodiazine skeleton the photoluminescent properties of the obtained compounds were measured in different solvents and are presented in Table 14.

While the partially saturated dihydro-pyrrolo-PY (**82a**–**d**) are red-shifted and have a poor or moderate quantum yield (around 5–40%), the tetrahydro-pyrrolodiazine (**81b**,**c**) have a negligible quantum yield (less than 5%). The fully aromatized pyrrolo-PY (**79a**–**c** and **80a**–**c**) are highly powerful blue emitters with extremely high quantum yields (up to 90%). The highly blue emitters of completely aromatized pyrrolo-PH (**79h** and **80g**–**i**) exhibit a low quantum yield of less than 10% and an unusual, blue-shifted absorption with λ_max_ of absorption around 314–322 nm.

Regarding the effect of the substituent from the 7th position of the pyrrolodiazine skeleton, the pyrrolo-PY compounds exhibit intense blue fluorescence and have a very high quantum yield when the substituent is an ester or amide group, whereas the quantum yield is negligibly low when the substituent is a ketone (see Figure 24).

We confirmed the experimental results through theoretical calculations into another study from our group [65]. According to computational study, the first nπ* state may be significantly stabilized when the carbonyl group from the 7th position of the pyrrolo-PY moiety is oriented so that it faces the diazine nitrogen. Given the lower (fluorescent) ππ* state expected for all conformations, such an effect does not seem to affect the fluorescence of the ester derivative. On the other hand, we found that the same nπ* state is predicted to have a lower energy than the first ππ* state for the most stable conformation of the benzoyl-substituted derivative, which is consistent with the weak fluorescence that was experimentally seen.

We demonstrate how the highly fluorescent ester-substituted pyrrolopyridazine’s solvent-dependent photophysical characteristics can be accurately predicted by TD-DFT in SS-PCM solvation when hybrid functionals such as B3LYP or PBE0 are used, together with a reasonable basis set.

In order to increase the fluorescent properties of the pyrrolo-PY derivatives, we studied the influence of the substituent from phenyl ring on the 2nd position of pyrrolo-PY skeleton [66], and used the strategies adopted for construction of fluorescent pyrrolo-PY derivatives are similar to those presented in the previous study [64] (see Figure 25).

The reactions under MW irradiation have shown some important advantages such as slightly higher yields and decreasing of the reaction time from 2 h to 5 min in liquid phase and 15 minutes in solid phase.

For the obtained compounds **86** and **87**, we investigated the photoluminescent properties (see Table 15).

The pyrrolo-PY substituents had some effect on the absorbance and fluorescence properties. Thus, the cycloadducts with carboethoxy groups, in the seventh position have a more fluorescence than those with a carbomethoxy groups, in the same position and also the cycloadducts with a *para*-chlorophenyl substituent in the second position of pyrrolo-PY moiety have a more fluorescence than those *para*-bromophenyl substituted.

Continuing our studies [67], we synthesized a new class of fused pyrrolodiazines **91**–**95** in order to obtain new fluorescent compounds by the same strategy-obtaining of the N-ylides fallowed by their cycloaddition with activated alkynes (see Figure 26). The reactions were conducted both under conventional TH and MW irradiation. The reactions under MW irradiation have shown a series of advantages such as decreasing the reaction time from 6 hours to 10 minutes and slightly higher yields.

After the synthesis of the desired pyrrolodiazines, we investigated their photoluminescent properties in non-polar solvents cyclohexane and dichloromethane. The measured parameters are presented in Table 16.

A relationship between structure of cycloadducts and fluorescence quantum yields was founded. Thus, the compounds with pyrrolo-pyridazine skeleton showed good quantum yield (around 25%), while compounds with pyrrolophthalazine skeleton showed a lower quantum yield.

In the next step we obtained some new bromoderivatives with increased reactivity by bromination of fluorescent pyrrolodiazines in heterogeneous catalysis in the presence of copper (II) bromide (see Figure 27). These bromo-derivatives can fluorescently label biological macromolecules due to their increased reactivity, being easily incorporated in those bio-macromolecules.

Having in view the results of the previous studies, we prepared a new family of pyrrolobenzo[*f*]quinoline derivatives [68] in order to obtain blue fluorescent azaheterocyclic derivatives. The pyrrolobenzo[*f*]quinolines **102a**–**c**, **103a**–**c** and **104c** were obtained by the same setup. In the first stage, benzo[*f*]quinolinium salts **100a**–**c** were obtained by quaternization of benzo[*f*]quinoline with bromoketones. In the 2nd stage, a typical Huisgen [3+2] dipolar cycloaddition of benzo[*f*]quinolinium ylides **101a**–**c** to the alkyne as dipolarophiles was performed (see Figure 28). A comparative study of these reactions was carried out under conventional thermal heating and under ultrasound (US) irradiation. The reactions under US irradiation have shown a series of advantages such as decreasing the reaction time from 2 days to 2 hours and slightly higher yields.

The photophysical properties of azatetracyclic derivatives **102a**–**c**, **103a**–**c** and **104c** were studied in cyclohexane and trichloromethane, respectively, and are presented in Table 17.

The obtained cycloadducts are blue emitters having λ_max_ of fluorescence around 430–450 nm. The cycloadducts with similar structure **102a**,**b** and **103a**,**b** present small differences in their electronic spectra, while the cycloadducts with bulky pivaloyl group **102c** and **103c** or without keto group in the 3rd position **104c** shows more intense fluorescence emission.

As a continuation of this work, in another study [69] we derivatized the previously obtained cycloadducts **102a**,**b** and **103a**,**b** in order to use this blue fluorescent azaheterocyclic derivatives as fluorescent biomarkers. In the first step, we applied this functionalization to the tetracyclic cycloadduct products **102a**,**b** and **103a**,**b** by the bromination in heterogeneous catalysis obtaining a mixture of mono or dibrominated product type **105a**–**d** and **106a**–**d** (see Figure 29). This bromo-derivatives can fluorescently label biological macromolecules (peptides, proteins, DNA) due to their increased reactivity, being easily incorporated in those bio-macromolecules.

In the next step, by a nucleophilic substitution, the monobrominated cycloadducts **105a**–**d** were used as starting materials for the synthesis of corresponding azides **107a**–**d**. The reaction was carried out in tetrabutylammonium bromide (TBAB) as phase transfer catalyst and in a chloroform/water mixture (see Figure 29).

For the new fluorophores (bromides **105a**–**d** and azides **107a**–**d**) we investigated optical properties on chloroform diluted solutions (see Table 18).

When comparing the fluorescence intensity of the bromides **105a**–**d** to the fluorescence of the azides **107a**–**d**, we observed a decrease in fluorescence, due the alteration of the planar structure of the molecule. The same effect is highlighted when comparing the fluorescence intensity of the underivatized cycloadducts **102a**,**b** and **103a**,**b** with the fluorescence intensity of the derivatized azatetracyclic bromides **105a**–**d** or azides **107a**–**d**.

Another research field of interest to us, is of the compounds which contain a pyrrolo[2,1-*a*]isoquinoline or imidazo[2,1-*a*]isoquinoline framework, due to their potential biological activity, but also to their extended π-conjugated systems that make them excellent candidates for photophysical applications [70]. Thus, we synthesized several new pyrrolo [2,1-*a*]isoquinolines and imidazo[2,1-*a*]isoquinolines in order to realize a complete study regarding their photophysical properties and potential applications in the field.

The chosen method for the assembly of fused target polyheterocycles relied on 1,3-dipolar cycloaddition of different isoquinolinium ylides **111a**–**c** (in situ generated in basic medium from corresponding salts **110a**–**c**, which were obtained by isoquinoline **108** alkylation with halides **109a**–**c**) to ethyl propiolate or ethyl cyanoformate. The intermediate dihydropyrrolo[2,1-*a*]isoquinolines **112′a**–**c** underwent oxidative dehydrogenation under atmospheric conditions, yielding the final compounds **112a**–**c** in good yields (63–80%). Using ethyl cyanoformate as dipolarophile in similar conditions, we obtained imidazo[2,1-*a*]isoquinolines **113a**–**c** presumably via dihydroderivatives **113′a**–**c** (see Figure 30).

The electronic absorption and emission spectra of the obtained azaheterocycles were recorded in dichlorometane (DCM) and dimethylsulphoxide (DMSO), and are presented in Table 19. The substituents on the 1st and 3rd positions of the pyrrole ring (COOMe, COOEt) have a low influence on the spectral pattern of pyrroloisoquinolines **112b** and **112c**. In addition, the solvent polarities have little to no influence on the electronic absorption spectra of pyrrolo- and imidazo-isoquinolines, suggesting that the ground state of these derivatives is not influenced by the solvent polarity. The extended π-π* conjugation in the imidazoisoquinolines system due to the substituents from the 1st and 3rd positions of the pyrrole ring induced a bathochromic shift of the absorption maxima. In DMSO, in the case of pyrroloisoquinolines, a hypsochromic shift of the absorption band occurs.

In the case of the compound **112a** containing CN group on the pyrrole ring, a similar behaviour of the emission spectra to absorption spectra was observed, namely a hypsochromic shift in both DCM and DMSO solvents, and no influence of the solvent polarity on the position of the emission maxima. The substituents on the pyrrole ring have no influence on the position of emission bands for derivatives **113a**–**c**, excepting on the compound **113a** which displays a hypsochromic shift in dichloromethane. The emission bands of imidazoisoquinolines **113a**–**c** are hypsochromic shifted in DCM and DMSO comparing with the emission bands of pyrroloisoquinoline derivatives **112a**–**c**. Time-correlated single photon counting (TCSPC) technique was used to determine the luminescence lifetimes. In DMSO, over the entire emission range, a double-exponential function describes better the emission decays for all investigated compounds. In DMSO, pyrroloisoquinolines **112a**–**c** have longer fluorescence lifetime τ1 than those of imidazoisoquinolines **113a**–**c**. The lifetimes of the excited-state for all compounds are in the nanosecond timescale.

Indolizine derivatives with phenanthroline skeleton [71,72,73] were another research field in our group due to both their potential biological activities and their extended π-electron system which make them valuable materials in the construction of new optoelectronic devices. Compounds **114a**–**g** and **115a**–**e** were synthesized in our group using the same 3+2 dipolar cycloaddition strategy of cycloimmonium ylides to activated alkynes (see Figure 31) [71]. In this case 4,7-phenanthrolin-4-ium and 1,7-phenanthrolin-7-ium ylides, generated from the corresponding monoquaternary **116** and **117** salts, were added to ethyl propiolate or dimethyl acetylene dicarboxylate DMAD.

The absorption spectra of all the compounds present the fine structure specific to phenanthrene spectrum. The extended π-system of the 1,7-phenanthroline compounds explained the bathochromic shift to longer wavelength absorption band as compared to the UV-Vis absorption spectra of 4,7-phenanthroline derivatives. The position of the absorption and emission maxima of phenanthroline derivatives are highly influenced by the substituents on the pyrrole ring, as can be observed from Table 20. The introduction of CN, COOMe and COOEt groups in the 9th and 7th positions of the pyrrole ring determine a hypsochromic shift of the absorption and emission maxima of phenanthroline derivatives (**114a**, **114b**, **115a**). The same effect on the absorption and fluorescence spectra is obtained when a third substituent (COOMe) is introduced in the 8th position of the pyrrole ring, only in the case of the compound **114e**.

The phenanthroline derivatives displayed broad and structureless emission band in 425 to 480 nm depending on the substituent nature at the pyrrole moiety. For 1,7-phenanthroline derivatives the emission band vas found in the 440–450 nm region, while for 4,7-phenanthrolines the emission band was found in 425–480 nm region. The position and the nature of the substituent at the pyrrole ring have a great influence on the fluorescence quantum yield of pyrrolo-phenanthroline derivatives. Figure 32 illustrates the effect of substituents from the pyrrole ring on the fluorescence yield of pyrrolophenanthroline derivatives.

The fluorescence emission of disubstituted phenanthroline derivatives (**114d**, **115c**, **115d**) is practically quenched by the presence of halogens (Cl, Br) in the para position of phenacyl group (COC_6_H_4_) from 9th position of the pyrrole ring. The emission quantum yield decreases significantly (**115e**−0.044) when COC_6_H_4_OMe-(*p*) group is introduced in the 9th position of the pyrrole ring due to the withdrawing effect of the methoxy group which determines a decrease in conjugation in the 1,7-phenanthroline system. Time-correlated single photon counting method in dichloromethane was used for the fluorescence lifetimes (τ) estimation.

The obtaining of small organic molecules capable to be incorporated in specific biomolecules or used in biomedical optical imaging was another research field in our group. A promising scaffold able to fulfil this purpose could be the bipyridyl. Bipyridyl, having two heterocyclic cores, can involve one or both of them in the formation of ylides and/or cycloaddition products. Thus, for these compounds we can obtain mono-salts [74], mono-ylides [74], mono-cycloadducts (indolizines) [74], mono-indolizine mono-salts [75,76,77], mono-indolizine mono-ylides [75,76,77], and theoretically bis-indolizines.

In one of our studies [74] we report on the synthesis, fluorescence properties and a preliminary evaluation of pyridyl-indolizine containing anthracene moiety as a DNA binding agent. The conversion of the ethyl ester group from the 1st position of the indolizine into the corresponding propargyl-based indolizine derivative facilitate the addition of anthracenyl group through a “*click*” reaction. Fluorescent pyridyl-indolizine derivative **124** was synthesized in moderate yield in an adapted four step synthesis. In the first step a cycloaddition of 4,4′-dipyridinium ylide, generated from the corresponding monoquaternary 4,4′-dipyridinium phenacyl salt **118**, with ethyl propiolate as activated alkyne gives an indolizine intermediate **119**. Isolated indolizine **119** ester was hydrolysed into the corresponding carboxylic acid derivative **120** in the second step, then in the third step was reacted with propargyl amine to yield alkyne-substituted indolizine derivative **121**, suitable for further “*click*” reactions. Separately, chloromethylanthracene **122** was straightforwardly transformed into corresponding azide **123**. In the final step, alkyne indolizine **121** was reacted together with azide **123** in a “*click*” type reaction to yield the final substituted pyridyl-indolizine **124** (see Figure 33).

The emission properties of compounds **121** and **124** were investigated in DMF at different pH values using 1xTAE buffer solutions (40 mM Tris, 20 mM acetic acid and 1 mM EDTA), and are presented in Table 21. The excitation wavelength was 395 nm and the emission spectra were recorded in 425–700 nm domain.

The emission of precursor alkyne **121** at acidic pH value of 2.0 is 6-fold stronger than at pH values of 5.0–12.0, while the fluorescence intensity of compound **124** at pH = 12.0 is 7-fold greater than at pH = 5.0. In the case of the precursor alkyne **121**, the emission spectrum presents a band with a shoulder at 495 nm at pH = 12.0 and a strong bathochromic shift of the emission maximum at pH values from 8.0 to 5.0, while the emission spectrum of compound **124** at pH = 5.0–12.0 presents bands with a shoulder at 475 nm. At pH = 12.0, the solution emits strongly in the green spectral window at 475 nm.

Agarose gel electrophoresis analysis and spectroscopic investigations were performed in order to investigate the interaction of nucleic acids with pyridine-indolizines **121** and **124**. Thus, deoxyribonucleic acid, low molecular weight from salmon sperm (sDNA) was used as a natural double-stranded DNA to test the binding properties of these compounds by agarose gel electrophoresis. The investigation shows that both investigated compounds **121** and **124** interact with sDNA.

The interactions of compounds **121** and **124** with sDNA was examined also by UV-visible and fluorescence investigation. Absorption and emission spectra were recorded before the addition of sDNA, immediately after the addition of sDNA solution and after the incubation of the indolizine-sDNA mixture at room temperature for 24 h. The absorption spectrum of compound **121** changes only after 24 h by the decrease of the absorption band intensity. Contrarily the absorption spectrum of compound **124** showed instantaneous changes in the shape of the band, upon the addition of sDNA, the structured band yielded by anthracene suffering modifications and after 24 h transforming into a broad band with a slightly weaker intensity. Thus, the presence of the anthracene moiety in the structure of indolizine **124** facilitates easier interaction with nucleic acids due to its higher affinity for DNA when compared to the propargyl moiety of the indolizine **121**. The same conclusion is obtained after the study of the fluorescence spectra of the mixtures: in case of compound **121**, alteration of the initial shape or intensity of the emission band was only observed after addition of sDNA and 24 h of incubation, while for the compound **124** immediate increase in fluorescence intensity of the mixture was observed, followed by additional increase after 24 h.

In order to investigate the ability of mono-indolizine mono-salts to interact with DNA, we used spectral methods UV-vis and fluorescence spectrometry [75]. Thus, a series of mono-indolizine mono-salts **125a**–**e** were synthesized starting from the corresponding mono-indolizines **126** by alkylating with halogeno-ketones **127**. The monoindolizines **126** were obtained by the cycloaddition of the corresponding ylides, generated in situ from the mono salts **128**, with ethyl propiolate (see Figure 34).

The absorption spectra of the mono-indolizine mono-salts were recorded in aqueous solution. The electron-donating methyl or methoxy group from the last inserted aromatic ring leads to a small bathochromic shift. DNA solution was progressively added to **125a** and the spectra were recorded, in order to study the interaction of compound **125a** with DNA. The hypsochromic shift of dye absorption maxima with the progressive addition of the DNA solution reflect the DNA binding to **125a**. Moreover, in the UV region (350–360 nm) a hyperchromic effect was observed. This typical hyperchromic effect is caused by the damaging of the DNA double-helix structure due to the intercalation of **125a** to DNA.

The interaction of **125a** with DNA was also investigated by emission spectra in acidic conditions since our investigated dyes display a lower solubility and instability in weakly basic condition. The fluorescence reached a maximum in pH range 1.8–3.6. The fluorescence intensities vary as follow: **125b** > **125a** > **125e** or **125d** > **125c**, being relatively higher in pH range 2.5–4.5. At higher pH values the solubility of dyes was very low due to the ylide formation.

Some preliminary investigation used herring sperm DNA (hsDNA) or control plasmid pUC19 in order to establish the interaction of the dyes with DNA. The fluorescence quenching of **125a** upon addition of hsDNA shows that the dye **125a** interact with DNA. The fluorescence of **125a-DNA** system was investigated at different dye (**125a**) concentration and quenching extent has a maximum value at 6 μM concentration of dye. Supplementary investigation on the interaction mechanism for binding of **125a** to hsDNA showed a similarly interaction way of **125a** and ethidium bromide with hsDNA, compound **125a** being binded in the minor groove of DNA.

Since the solubility the compound **125a** is poor, to increase its solubility and to extend its applications as cell staining agent or cell pH sensitive dye, we proposed the incorporation of compound **125a** in β-cyclodextrin (β-CD) [76]. The reversible transformation of the pyridinium moiety in compound **125a** to the corresponding nitrogen ylide **129a** under proper pH condition (see Figure 35), influence its fluorescence emission spectra, making it a pH sensible fluorescent dye.

The inclusion complex of indolizinyl-pyridinium salt in β-cyclodextrin (β-CD) was prepared by heating the equimolar amounts of components in water till 110 °C for 60 min, cooling down the solution under stirring for 6 h to reach the equilibrium followed by filtration through Phenex syringe filters (pore size: 0.45 μm).

The ESI-MS experiments and molecular docking studies confirmed the formation of an inclusion complex between indolizine derivative and β-cyclodextrin in 1:1 and 1:2 ratios. The cytotoxicity of the inclusion complex was considerably reduced comparing with the cytotoxicity of free indolizine on both HeLa (human cervix adenocarcinoma) and NHDF (normal human dermal fibroblasts) cells.

For the first time we demonstrated that the toxicity of a fluorescent dye was strongly reduced by the formation of cyclodextrin inclusion complex, allowing the successful application in cell staining. The study regarding cell membrane permeability showed that the nontoxic inclusion complexes mixture specifically accumulates in cell acidic organelles since could not pass through the cell plasma membrane.

Recent, ref. [77] we prepared three new cyclodextrin encapsulated pH sensitive dyes and investigated their ability for self-aggregation and in vitro assessments as fluorescent cellular probes. The mono-indolizine mono-salts **130a**–**c** (their structures are presented in Figure 36) were synthesized in moderate yields in an adopted two step strategy based on our methodological background.

Due to poor solubility in water of the indolizinyl-pyridinium salts **130a**–**c**, these compounds were tested for the formation of inclusion complexes with β-CD. We started with an equivalent of each compound suspended in water, followed by the addition of an excess of β-CD (5 eq.) and heating to 90 °C until the reaction solution became transparent (after 25 min of heating). In case of **130c_CD** the solutions remaining completely transparent after cooling down, while in case of **130a_CD** and **130b_CD** the formation of slightly cloudy solutions was observed. The excess up to 10 equivalents of the CD amount in the reaction mixture still did not change the appearance for the **130a_CD** and **130b_CD** at room temperature. These solutions were used in the subsequent analyses only after a microfiltration procedure.

The ESI-MS experiments and molecular docking studies confirmed a 1:1 and 1:2 ratios between indolizine derivative and β-cyclodextrin in the inclusion complex.

Absorption and emission spectra of indolizine derivatives **130a**–**c** and their inclusion complexes **130a**–**c_CD** were recorded at 1.0, 7.4, and 13.0 pH values. (1.0, 7.4, and 13.0) and the results compared. The indolizines **130a** and **130b** have similar absorption spectra, but slightly different than the absorption spectra of the indolizine **130c**. At pH = 13.0, all the investigated indolizines have shown similar spectra. In the case of inclusion complexes **130a**–**c_CD**, their absorption spectra at acidic and neutral pH values (1.0 and 7.4, respectively) are similar for all the investigated complexes, but different compared to the absorption spectra of the starting indolizines **130a**–**c**. At basic pH values (pH = 13.0), the shape of the spectra of all complexes **130a**–**c_CD** is drastically changed in comparison to each other and to the starting indolizines **130a**–**c**.

The excitation wavelength was 420 nm when fluorescence spectra of **130a**–**c** and **130a**–**c_CD** were measured at different pH values. Starting indolizines **130a**–**c** exhibited similar emission spectra at acidic and neutral pH values, with an emission band around 550 nm. This band is approximately two times higher in intensity at acidic than neutral pH value. At basic pH values (pH = 13.0), the indolizines shown low to no fluorescence. The reversible transformation of pyridinium salts into analogous non-fluorescent ylides may explain this pH-dependent behaviour. The inclusion complexes **130a**–**c_CD** demonstrated slightly different pH dependent behaviours, showing a similar low intensity at basic pH values, but different intensities at acidic pH values. The fluorescence intensities of the inclusion complexes **130a_CD** and **130b_CD** at acidic pH are only slightly higher than intensities at neutral pH values, while in the case of the inclusion complex **130c_CD**, the fluorescence intensities at both the acidic and neutral pH were comparable. The formation of the inclusion complexes altered the planar structure of the indolizinyl-pyridinium salt molecules due to the specific rotation limitations induced by the steric hindrances with the CD, which determine a decrease of fluorescence intensity of the inclusion complexes at acidic pH values.

Supplementary studies shown that the inclusion complexes have no cytotoxicity, they specifically accumulate within acidic organelles or mitochondria due to cellular permeability, and their intracellular fluorescence increase over a 24 hours period with outstanding signal stability.

## 3. Concluding Remarks

In conclusion, the present review article deals with design, synthesis, and photophysical properties of azaheterocycle-based materials. Several synthetic methodologies were used for the obtaining of fluorescent 5, 6 membered and fused azaheterocycles. The structure-fluorescence relationship of the obtained compound was investigated and allowed the authors to obtain compounds with better fluorescent properties in terms of emission wavelength, emission intensity and quantum yield.

The presented approaches regarding the synthesis of fluorescent azaheterocycles, using varied methodologies, enable researchers to create a library of multifunctional derivatives, possessing high efficiency of fluorescence.

The tunable emission properties of the azaheterocyclic compound make them valuable materials as potential luminophores in OLED devices, as luminescent solar concentrators or scintillators, as fluorometric (naked-eye chemosensor) molecular sensors for detection of metal ions, strong acids and bases with high selectivity, as thermally activated delayed fluorescence emitters and as fluorescent probe.

## Data Availability

Not applicable.

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
