# Peer review of "A Review on the Synthesis of Fluorescent Five- and Six-Membered Ring Azaheterocycles"

_molecules, 2022, doi:10.3390/molecules27196321_

Round 1
Reviewer 1 Report
The Review “A Review on the Synthesis of Fluorescent Five- and Six- Membered Ring Azaheterocycles” describes what the Azaheterocycles ring with five and six members are important tools for the obtaining of fluorescent materials and fluorescent sensors. In addition, it refers the relevant advances in the synthesis of azahetero-cyclic derivatives and their optical properties investigation, in the last ten years.
In my opinion the Review can be published in Molecules after the authors answer the following questions.
- The Introduction and Conclusions should be expanded as in the present form they no reflect all the good work cited by the authors
- In some parts of the work -for example in Table 1. Fluorescent properties of the selected 5aa and 5ba in DMSO- the authors must explain why they used these compounds to determine their properties.
In summary, my main observation regarding this good complilation work lies in the way in which the results are presented, which was confused somewhere -for example, theTable 10. Spectroscopic behavior of dyes A2 and D1 in different solvents, which it is not sufficiently explained-.
I considered that the Ms. can be published with minor corrections
Author Response
The Review “A Review on the Synthesis of Fluorescent Five- and Six- Membered Ring Azaheterocycles” describes what the Azaheterocycles ring with five and six members are important tools for the obtaining of fluorescent materials and fluorescent sensors. In addition, it refers the relevant advances in the synthesis of azahetero-cyclic derivatives and their optical properties investigation, in the last ten years.
In my opinion the Review can be published in Molecules after the authors answer the following questions.
- The Introduction and Conclusions should be expanded as in the present form they no reflect all the good work cited by the authors
We have modified the Introduction and Conclusion chapters.
- In some parts of the work -for example in Table 1. Fluorescent properties of the selected 5aa and 5ba in DMSO- the authors must explain why they used these compounds to determine their properties.
In this particular case, the authors did not explain why they used these compounds to determine their fluorescent properties, but most probably were selected only the compound that present fluorescence.
In summary, my main observation regarding this good complilation work lies in the way in which the results are presented, which was confused somewhere -for example, theTable 10. Spectroscopic behavior of dyes A2 and D1 in different solvents, which it is not sufficiently explained-.
We presented in Table 10 (Table 7 after review reorganization) the main optical parameters (absorption and emission wave, molar absorption coefficient, fluorescence yield and lifetime) A discussion on the influence of the solvent on these optical parameters was presented above Table 7.
I considered that the Ms. can be published with minor corrections
Thank you for your suggestions.
Reviewer 2 Report
This review summarizes some results on the synthesis five- and six-membered azaheterocycles with fluorescent properties including the authors’ own contribution. It could be of interest not only for synthetic chemists but also for chemists involved in material sciences. In my opinion, this work can be published in Molecules after the following points will be addressed.
Notes/remarks for the authors:
1) The organization of the review is somewhat confusing. For example, compound 16 is located in chapter Five-membered heterocycles but this compound consists of one 5- and two 6-membered heterocycles! Similar situation is with compounds 23, 66, 69. At the same time, in the third part, the authors’ contribution, heterocycles are not classified according to the size at all.
It would be better to divide all heterocycles to monocyclic and fused regardless of the ring size.
2) The topic of the review is very fruitful. Thera are a lot of works which are not covered for example RSC Adv., 2015,5, 94551-94561
3) Schemes numbering should be added
4) In top Scheme at page 6 an arrow to compounds 17 and 18 should be crossed out. These products were not formed.
5) In bottom Scheme at page 6 temperature should be presented in Celsius degrees.
6) Page 12 «caging applications» should be replaced with «cellular applications» and «cored» - with «fused»
7) In chemical names of fused heterocycles, a letter in brackets should be in italic style.
Author Response
This review summarizes some results on the synthesis five- and six-membered azaheterocycles with fluorescent properties including the authors’ own contribution. It could be of interest not only for synthetic chemists but also for chemists involved in material sciences. In my opinion, this work can be published in Molecules after the following points will be addressed.
Notes/remarks for the authors:
1) The organization of the review is somewhat confusing. For example, compound 16 is located in chapter Five-membered heterocycles but this compound consists of one 5- and two 6-membered heterocycles! Similar situation is with compounds 23, 66, 69. At the same time, in the third part, the authors’ contribution, heterocycles are not classified according to the size at all.
It would be better to divide all heterocycles to monocyclic and fused regardless of the ring size.
Thank you for your suggestions, we have reorganized the review accordingly.
2) The topic of the review is very fruitful. Thera are a lot of works which are not covered for example RSC Adv., 2015,5, 94551-94561
Thank you for your suggestions, we explained better in Introduction the methodology adopted for the literature search and selection, and inserted the indicated reference in the review.
3) Schemes numbering should be added
Thank you for your suggestions, we have inserted the number and title of the schemes, accordingly.
4) In top Scheme at page 6 an arrow to compounds 17 and 18 should be crossed out. These products were not formed.
Done, thank you for your suggestions.
5) In bottom Scheme at page 6 temperature should be presented in Celsius degrees.
Done, thank you for your suggestions.
6) Page 12 «caging applications» should be replaced with «cellular applications» and «cored» - with «fused»
Done, thank you for your suggestions.
7) In chemical names of fused heterocycles, a letter in brackets should be in italic style.
Done, thank you for your suggestions.
Round 2
Reviewer 2 Report
The first sentence of the Concluding Remarks is not correct. The authors did not review OLED devices. "the present review article deals with application of azaheterocycle-based materials in OLED devices, including design, synthesis, and photophysical properties of such materials." should be corrected like this: "the present review article deals with design, synthesis, and photophysical properties of azaheterocycle-based materials."
Author Response
We have done the correction.
Thank you for your suggestions.